# Technology Roadmap for Eco-Friendly Building Materials Industry

**Hyunsook Shim** [1] , **Taeyeon Kim** [2] **and Gyunghyun Choi** [3,*]

1   Pium Innovation co., Ltd., 37, Hwangsaeul-ro 258, Bundang-gu, Seongnam-si, Gyeonggi-do 13595, Korea; dodohyoun@hanmail.net
2   Department of Architectural Engineering, Yonsei University, Seoul 03722, Korea; tkim@yonsei.ac.kr
3   Graduate School of Technology and Innovation Management, Hanyang University, Seoul 04763, Korea
*   Correspondence: ghchoi@hanyang.ac.kr; Tel.: +82-2-2220-2250

**Abstract:** As quality of life has improved, the need for high-performance building materials that meet specific technological requirements has increased. Residential environments have also changed owing to climate change. A technology roadmap could define and systematically reflect a timeline for the development of future core technologies. The purpose of this research is to build a technology roadmap that could be utilized for the development of technology in the eco-friendly building material industry. This research is composed of multiple analysis processes—patent analysis, Delphi, and analytic hierarchy process analysis—that minimize the uncertainty caused by the lack of information in the eco-friendly construction industry by securing objective future forecast data. Subsequently, the quality function deployment test is implemented to verify the feasibility of the technology roadmap that is constructed. The design of various types of functional, low-carbon building materials could reduce carbon emissions and save energy by ensuring a hazardous-material-free market in the future. This design development roadmap is required to complement this technology roadmap.

**Keywords:** eco-friendly building materials; functional building materials; technology roadmap

## 1. Introduction

Climate change is gradually progressing, affecting all sectors of society. Therefore, consolidated and systematic measures to respond to climate change that can be applied in all sectors of society are required. Globally, climate change and energy problems have become a priority when establishing national policies, and the focus is on nurturing eco-friendly technologies and green energy businesses as major industries [1–3].

The United States of America is promoting policies such as the expansion of eco-housing and hybrid vehicle supplies. Furthermore, the European Union (EU) demands the control of greenhouse gas emissions through the entire process of product manufacturing, with plans to reduce greenhouse gases by 20% and to replace 20% of the total energy with renewable energy by the year 2020. Japan announced the 'New Growth Strategy' in 2011, with the objective of becoming an environmental and energy power, and is in the process of implementing its 4th basic plan of science and technology, which has 'green innovation' as a major driving force. Moreover, Japan has achieved globally superior technological levels in fields related to re-utilization and re-resourcing of waste [2].

China also decided to expand the renewable energy composition of their total energy requirements to 15% by 2020. To accomplish this goal, they established a plan to promote new industries with high potential for environmental preservation and new energy as part of its '12th 5-year development plan for science and technology', which was established in 2011 [2].

South Korea has also submitted a spontaneous reduction plan to reduce their projected basic greenhouse gas emission (business as usual) value to 37% (25.7% for domestic businesses, 11.3% for international businesses) by 2030 in accordance with the UN climate change convention (United Nations Framework on Climate Change). Changes in the environment due to climate change have initiated technological innovation in the environmental industry, as well as in other industries. Among them, eco-friendly technology faces the maximum social demand [3].

In the residential environment, the awareness level of eco-friendly approaches is becoming higher. Thus, the eco-friendly building material market size in South Korea has grown to $3.7 billion in 2015 from $2.1 billion in 2010, which is an annual growth of 15% on average; it is observable that the market demand for eco-friendly residential environments is ever growing. Accordingly, the government is trying to build a 'low-energy and eco-friendly residential environment' by means of the 'Green Building Certification System' and the 'Healthy and Environment-Friendly Housing Construction Standards'. These rapid changes in the external environment heighten the uncertainty of future markets, leading to difficulty in the establishment of technology development policies and their relevant strategies. Therefore, some industries utilize a technology roadmap that derives the core technology after selecting major technological items that must be developed in the future in order to determine the potential demand based on a forecast of future markets. Already, in many countries, the technology roadmap is being used as an important method to determine industry trends and to give directions for technological development at governmental levels [4–7].

However, in the case of the environmental industry, the utilization of a technology roadmap is not as frequently carried out as in other industries. This is because the environmental industry has more aspects related to social contribution, rather than profit making, prompting the governments of many countries to politically support technological development, also providing research funding at governmental level [8]. Although these government-led Research and Development (R&D) investments have the positive function of leading the environmental industry with consistency and stability, it occasionally becomes a factor in hindering the utilization and research of a technology roadmap, thus delaying the changes required owing to market demands. Moreover, the manufacturers of Korean eco-friendly building materials are deficient in terms of technological innovation capabilities that could enable a flexible reaction to changes in the environment of such technological developments. In particular, the eco-friendly building material sector has a structural characteristic that makes it difficult to comprehend the market needs that are fundamental for future forecasts, in that the purchasers (construction companies) and the users (inhabitants) are different. Because of such industrial and structural characteristics, R&D in the eco-friendly building material industry has been led by the government, rather than by the market, and the technology roadmap has been defined not by the demands of the market, but by the direction of government policies. However, with the improvement in quality of life and the growth in market requirements for such highly functional building materials that require specific technology, the creation of a technology roadmap that systematically reflects the definition of future core technologies and provides a developmental timeline is urgently required [9].

Consequently, the following research question was raised. Could a roadmap that achieves objectivity and validity be developed in the eco-friendly building material industry based on insufficient data and information on the market and technology? This study demonstrates the challenging work, as well as its results, of investigating this research question. Therefore, the purpose of this research is to develop the technology roadmap for the eco-friendly building material industry.

The structure of the study is as follows. Section 2 reviews the previous research on technology roadmaps through a literature review, and then examines the establishment of the technology roadmap in the environmentally friendly industry. In Section 3, we propose a multi-analysis process for patent analysis, as well as Delphi and Analytical Hierarchical Process (AHP) analysis, as a research method to complement uncertainty and to obtain objective technological data for the eco-friendly building materials industry. In Section 4, we present a step-by-step analysis of the multiple analysis process.

Based on the analysis results, we build a technology roadmap for the eco-friendly building materials industry. Furthermore, the Quality Function Deployment (QFD) test is run to verify the validity of the study results. Finally, we discuss some of the distinctions and points of originality of this research, including some limitations of research and future direction of research.

The technology roadmap developed in this research is expected to improve the technological competitiveness of small and medium companies in the concerned industries by proposing the development of technologies appropriate to the market needs in the future.

## 2. Literature Review

The technology roadmap is a process of deriving the core technology from a selection of major types of technologies that need to be developed in future to fulfill the future needs, which are determined based on a forecast of the future market [10]. It is called a technology roadmap because it draws those directional roads of policy and technology that need to be planned and implemented to achieve the core technology objective, similar to the road maps of a city being drawn.

The technology roadmap was first used in Motorola, in the USA, in the 1970s, and spread out afterward across various industries. Gerdsri [11] and Fenwick et al. [12] presented research related to the technology development envelope and proposed a technology roadmap methodology that combines technological strategy with business strategy; further, he also presented a new technology roadmap approach method combining marketing techniques with decision-making methods. Subsequently, he proposed practical guidelines for the purpose of appraising the management of changes in organizational structure and technology road mappings by building up technology roadmaps and integrating consistent strategic planning processes within the process of product development. Moreover, his research presented in 2013 proposed a professional network mapping method that offers implicit cooperation between the researcher and working-level people [13,14].

Technology road mapping involves researching the relationships between the evolving market and the market under development, product and technology, and provides a structured means of communication [11,15]. Daim et al. [16] and Kockan et al. [17] carried out research on the development of the most viable future power train system, among a variety of alternatives, to improve the efficiency of the power train system using the technology roadmap research method; moreover, after confirming through case studies the increase in the competitiveness of Ford Motors by applying such built up technology roadmaps, they proposed a method to examine the process of investment in the technology and its application programs.

Research on technology roadmaps is underway in the Industry 4.0 in recent years. Lu and Weng [18] predicted core technological trends by analyzing the correlation between core technologies, market maturity, and technology maturity in a technology roadmap study of the fourth industry, the smart manufacturing industry. Jeong [19] analyzed patent development patterns by using text mining and deep learning techniques for patent documents in the field of transparent display, autonomous driving automobiles, and artificial intelligence.

The technology roadmap is a very useful process for understanding technological trends and proposes a direction for technological development in all energy-related industry fields at government levels, globally. Gough et al. [4] provided the current status, potential, and overview of barriers in long- and short-term technology allocation by building up a technology roadmap of carbon collection and storage for the UK government. Furthermore, Kelly and Daim [5], Amer and Daim [6] Krystyna et al. [7] applied the technology roadmap as a research method in the government's new renewable energy areas, utilizing it in order saving costs and forecast potential growth trends. Recently, Ceci et al. [20] conducted a technology roadmap study on energy storage systems to reduce carbon emissions and increase renewable energy use. Additionally, Yu and Li [21] used the basic principles and methods of the technology roadmap in the analysis of the eco-friendly cultivation techniques for the plant extract industry.

Furthermore, the South Korean government carried out research in order to predict the impact on the environmental sectors of utilizing research, such as the scenario method, that could take into account future uncertainties in establishing national science and technology policies. However, the research so far has been predominantly limited to focusing on countermeasures or on technological change itself through the analysis and forecast of future trends. Specifically, in industries related to eco-friendly products, in which uncertainty and changeability levels are higher, roadmap research into future trends and the development of methodologies are currently insufficient.

## 3. Methods

This research intends to build up a technology roadmap related, among the building/construction fields, to the eco-friendly building material industry, which is currently facing increasing market demands following recent climate changes.

Normally, the most frequently used method for understanding the technological trends in order to build a technology roadmap is the questionnaire survey conducted on users. By using questionnaire surveys, industrial bodies can understand technological trends and develop the corresponding technology and expand their patents. Thus, future technologies could be predicted by understanding core technological keywords that reflect the market trends through patent analyses.

When building up a technology roadmap at the governmental level, the expert Delphi method is used most frequently as an important means of understanding technological trends and presenting the direction of technological development. Krystyna et al. [7] applied the technology roadmap research method utilizing the Delphi method in the Polish government's research forecasting the cost saving and potential growth trend in the new renewable energy sector.

In the eco-friendly building material industry, there is an absolute deficiency of related data or technological information that could be used as base materials to forecast the future. In addition, the fact that the users (inhabitants) and the purchasers (construction companies) are not identical presents a limitation in determining the technological needs of the market by either general questionnaire survey method or patent analysis alone.

### 3.1. Research Model

In this section, we propose our research model for building up the required roadmap. This model is composed of a quantitative-qualitative multiple analysis process in order to minimize the uncertainty and obtain objective future forecasting data that can rectify the information deficiency in the eco-friendly building material industry. In addition, this process could verify the feasibility of the technology roadmap by applying the QFD test method, which can measure in detail the needs of users and the order of priority for technological development. Figure 1 is the multiple analysis process model for this research.

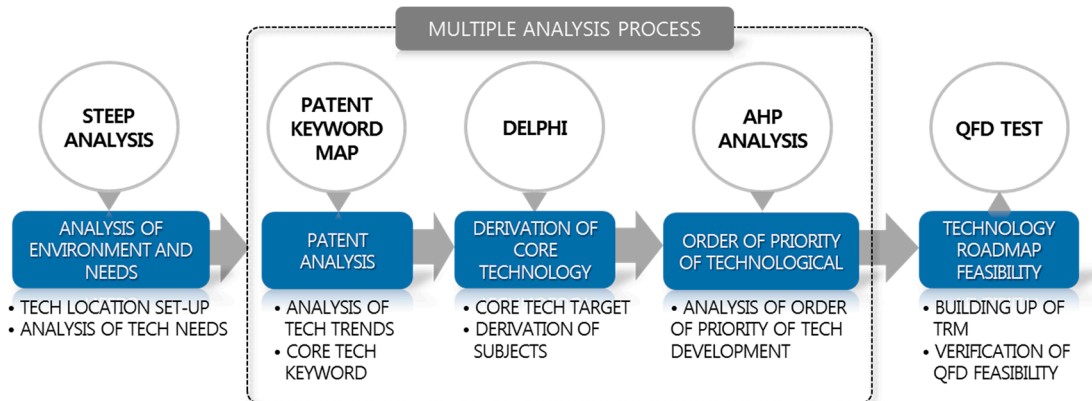

**Figure 1.** Proposed research model for technology roadmap.

The multiple analysis process illustrated in Figure 1 is composed of three phases—patent analysis, Delphi, and AHP—and is capable of building up data of higher objectivity by including all the quantitative-qualitative analyses. Subsequently, a QFD test is conducted for the feasibility verification of the constructed technology roadmap.

### 3.1.1. Patent Analysis

Patent analysis was performed using a quantitative method to comprehend the technological trends that contribute more objectively to the technological forecast scenarios for eco-friendly building materials. In terms of patent analysis programs, Wizdomain DB and BIZ IP analytical tools were utilized three times for surveys. After setting up the search ranges in the major, middle, and minor categories, the technological trends were analyzed by a keyword search; after removing noise via patent DB search and screening, the keywords for core technology were derived. Furthermore, based on the keyword map, the correlation among the keywords for core technology was analyzed to forecast future technologies. The patents derived through the searches in the major, middle, and minor categories were screened 10 times per range to eliminate those not patented as building materials. All registered or filed patents were included, but those rejected were excluded.

### 3.1.2. Delphi

The Delphi technique was implemented to predict the target performance index of the core technology for the technological keywords obtained through patent analysis. The eco-friendly building material industry, which was the target of this research, is a typical government-led technological development industry in which the direction of technological development is led by governmental policies. Therefore, the degree of participation by policy makers, who are the interested parties in the establishment of the target performance index, was important [22]. Considering the industry characteristics, the Delphi method was implemented by involving policy experts from the Ministry of Environment and the Ministry of Land, Infrastructure and Transport. This technique was intended to derive the future forecasts for the eco-friendly building material technology roadmap with a greater degree of objectivity based on the opinions of experts.

In this research, the Delphi method was implemented, involving policy experts of the Ministry of Environment, the government authority concerned with the $CO_2$ emission reduction policy, and of the Ministry of Land, Infrastructure, and Transport, who are interested in environmental changes of the markets such as the enactment of new regulations related to the functional building materials, etc. The first implementation of Delphi was conducted using a face-to-face focus group interview with 12 experts concerned with eco-friendly building material policies among a total of 19 people, i.e., the sum of the 8 participants from the Ministry of Environment expert commission and the 11 participants from the Ministry of Land, Infrastructure, and Transport. The interview was conducted in their National R&D meeting, convened on October 2nd. The second implementation of Delphi was carried out simultaneously using telephone and email communication methods, and their opinions were collected during the third implementation of Delphi after screening.

### 3.1.3. AHP Analysis

After setting up the target performance index of core technology through Delphi implementation, AHP analysis was performed to grasp the importance for each item of the eco-friendly building material future technology and to derive the order of priority in the development of technology.

AHP analysis is a multi-standard decision-making method providing more systematic evaluation in cases where there are multiple evaluation standards and alternatives with mutually complex impacts affecting decision making, and for cases of qualitative elements for which quantitative measurement against an evaluation standard is difficult. It can provide systematic understanding of complex problems by simplifying them by positioning the evaluation standards that affect the decision making into a phasic hierarchy. Moreover, this process can calculate the degree of importance of the

constituent elements of evaluation standards considering not only their mutual interaction, but also their overall relationships.

The AHP analysis is at a very important stage on the timeline of the technological development phase of the market, product, and technology, in terms of achieving the core technology target performance of future technologies in the eco-friendly building material industry. Therefore, it was processed with more care to enhance the reliability of the whole technology roadmap and for securing the consistency. The AHP analysis questionnaire survey was carried out over 20 days, and the targets of the questionnaire survey were 37 people who participated in the forum for the housing regulations proposed for 'Construction Standard of the Clean Healthy Housing', namely, policy experts from the Ministry of Land, Infrastructure and Transport, the material handling staff of construction companies, designers, the research people of the companies producing building materials, etc. The participants took part in the survey predominantly through the face-to-face method, along with e-mail communication.

### 3.2. Feasibility Verification of Roadmaps

Once the 'technology roadmap' of the eco-friendly building material industry was built up through the three phased Patent analysis–Delphi–AHP multiple analysis process, a QFD test was performed as a verification for the purpose of supplementing the deficient parts of market survey information of the consumer's needs.

QFD is a quality management technique developed for the rapid development of products or services that accurately reflect the clients' needs [23]. As new technology products that were non-existent before lack previous patent or technical data, and because there is a limit to predicting future technology solely based on the opinions of professionals, the QFD test is occasionally applied from a measurable quantitative perspective in order to obtain the required quality of products [24]. Phaal et al. [15] also proposed the QFD test as a segmented method of building a technology roadmap after assessing the feasibility of measuring the order of priority of the users' needs and technological development in detail.

The QFD test verification stage is a differentiated technology roadmap creation method used in this research, and is a verification technique for determining the objectivity and the feasibility of constructing a technology roadmap for small and medium sized companies of building materials that rectifies the issues of information deficiency in the future forecast data and the market needs of this industry, and is designed with the participation of policy experts in consideration of the government-led characteristic of the R&D process.

In this research, all the consumers and suppliers of eco-friendly building materials are defined as clients, and the QFD test was processed against 15 construction and design companies, being the consumers, and 15 suppliers of materials.

## 4. Technology Roadmap for the Eco-Friendly Building Material Industry

### 4.1. Composition of the Roadmap

In general, an 'eco-friendly building material' is a product that can contribute to the saving of energy and resources, and can reduce environmental contamination. On the other hand, 'functional building material' refers to building materials with effective functions for the improvement of indoor atmospheric environments while satisfying the additional performance evaluation standard for adsorption, moisture absorption and desorption, and antibacterial and antifungal qualities, as well as satisfying the basic criteria of eco-friendliness by releasing limited amounts of hazardous substances [25]. As a countermeasure against climate change, efforts to save energy and reduce carbon emissions are being continuously exerted globally; further, there is a growing commercial demand for facilities to control the indoor air quality of residential environments. Consequently, these low-carbon building materials and functional building materials may be referred to as next-generation eco-friendly

building materials [26]. Therefore, this research will include 'low-carbon building materials' and 'functional building materials' as target technologies for the next generation of 'eco-friendly building materials'.

This research built up a technology roadmap by developing the three-phased multiple analysis process, comprising Patent analysis–Delphi–AHP—designed with the participation of policy experts in consideration of the government-led characteristic of R&D, while complementing the problems of eco-friendly construction industries which conspicuously lack objective data and information for the prediction of the future.

The Patent analysis, which is the first phase of the proposed method, was conducted using the Wizdomain DB and BIZ IP analytical tools in three steps, as described in 3.1.1. The number of patent applications was retrieved using the keywords 'technology for the removal of indoor pollutants' and 'building materials exhibiting low emission of pollutants' in both South Korea and Japan for the 17-year period from 2001 to 2017 in order to compare the patent trends related to 'low-carbon building materials' in South Korea and Japan. The patent application trends of technology related to low-carbon building materials are illustrated in Figure 2.

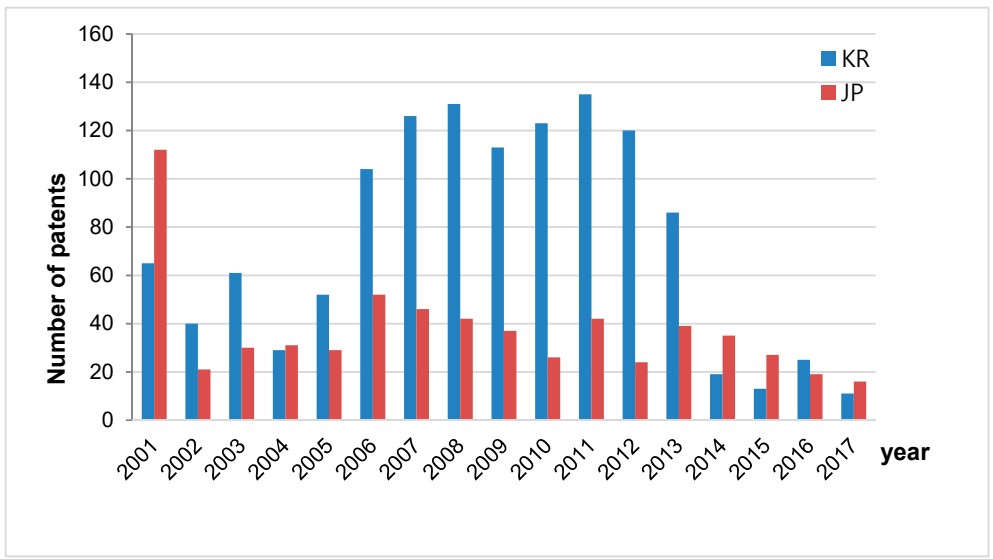

**Figure 2.** Application trends of low-carbon building materials in South Korea and Japan.

From Figure 2, the annual average numbers of patents are 49, 110, and 17 for the periods 2001–2004, 2005–2013, and 2014–2017, respectively. This seems to be due to related systems and legislation that took effect in 2003 and 2004. Since 2014, many companies have shifted the direction of their research into functional materials, resulting in very low patent activity.

To compare the patent application trends related to the 'functional building material' in South Korea and Japan, the patent applications were analyzed using the keyword 'technology related to moisture absorption and desorption'. The trend results of patent applications related to functional building material technology are as illustrated in Figure 3.

Also, from Figure 3, it can be observed that the average annual numbers of patents are 14, 33, and 20 for the periods 2001–2004, 2005–2013, and 2014–2017, respectively. It seemed that the average annual number of patent registrations increased during the period 2005–2013, as the government encouraged R&D in functional building materials through relevant regulations. However, the average number of patents for the period 2014–2017 was reduced, as the products were applied to the field. On the other hand, in Japan, where there are no separate regulations for functional building materials, the number of patents did not change significantly. As a result of the patent analysis, we observed that the number of patent registrations was related to the implementation of new institutions and to market demands.

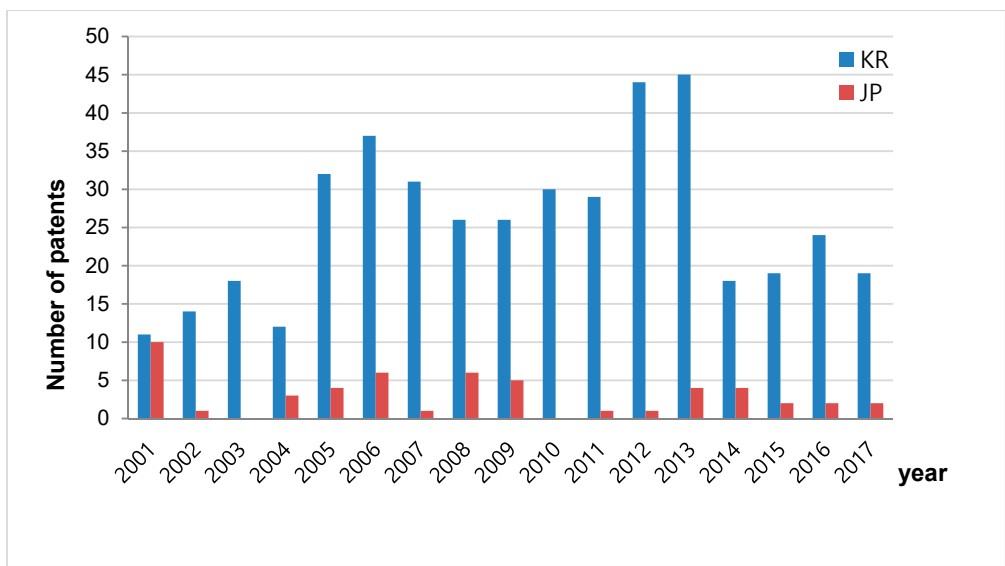

**Figure 3.** Trends of patent applications for functional building materials in South Korea and Japan.

Based on the patent analysis performed for the purpose of core technology keyword analysis, the 'building material patent keyword map' was analyzed. Table 1 lists the results of the derivation of core technology keywords following the patent keyword map analysis.

**Table 1.** Result of analysis of patent keyword map.

| Technology Trend | Patent Keyword Map | Analysis Result |
|---|---|---|
| Building material | - Eco-friendly, Sick House Syndrome, Indoor Air, Formaldehyde, High Functionality, Fungi<br>- Adsorption Substance, Diatomite, Natural Minerals, Minerals, Silicate, Zeolite | - Low-Carbon Building materials<br>- Functional Building materials |
| Adsorption Functional Building materials | - Porosity, Structure, Surface, Heat Treatment, Washing, Dispersion | - Performance manifested by processing techniques of raw materials rather than the raw material ingredients |
| Moisture Adsorption and Desorption Functional Building materials | - Zeolite, Diatomite –> Raw Materials<br>- Porosity, Plasticity –> Manufacturing Method | - Raw material processing techniques affect the performance considerably as well as the important raw material ingredients |

Expert Delphi was implemented as the second phase of the multiple analysis process to establish the target performance index of core technology based on the technology keywords obtained through patent analysis. Expert Delphi is composed of two alternatives—'low-carbon building material' and 'functional building material'—to minimize the risk of future forecasts. Figure 4 shows the multiple scenarios composed for the purpose of questionnaire survey for the eco-friendly building material technology roadmap target performance index and for establishing the hierarchy for the purpose of needs analysis.

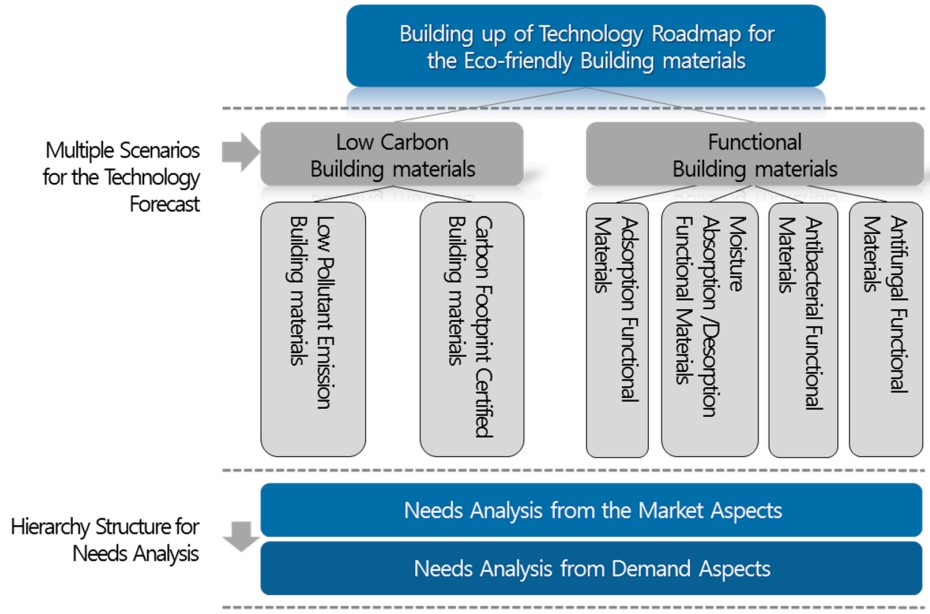

**Figure 4.** Hierarchy of needs analysis for eco-friendly building materials.

After setting up the target performance index for the core technologies in the eco-friendly building material industry through the implementation of Delphi, the AHP, the 3rd phase of multiple analysis process, was performed in order to understand the importance of future technologies for each function and to derive the order of priority in technology development. Figure 5 illustrates an analysis hierarchy model for AHP, which structures the questionnaire items for technology development into two stages.

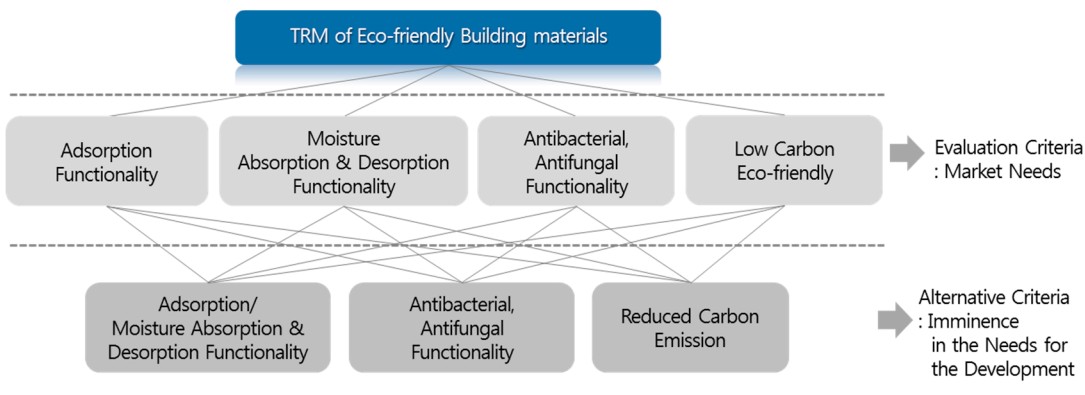

**Figure 5.** AHP analysis hierarchy model structure.

The AHP analyzed the items for the analysis of 'market needs' in two categories based on their importance according to the duality comparison between the items and 'the imminence of the development', and carried out analysis of the order of priority of technology development. Table 2 lists the results of the AHP.

Based on the analysis of core technology importance as listed in Table 2, building materials with moisture absorption and desorption functionality, which demonstrate the capacity to control the humidity, had the highest degree of importance among all the core technology subjects obtained based on the evaluation standard items of the first phase of the hierarchy. The analysis of the order of priority for technological development demonstrated that the order was adsorption, moisture absorption and desorption functional building material, antibacterial and antifungal functional building materials, and low-carbon eco-friendly building materials, in terms of their respective imminence of development.

**Table 2.** AHP results.

| Items | Result of Analysis of the Importance of Core Technology | Result of the Analysis of the Order of Priority in the Technology Development |
|---|---|---|
| Low-Carbon Building Materials | 0.02 | 0.06 |
| Adsorption Functionality | 0.11 | 0.71 |
| Moisture Absorption and Desorption Functionality | 0.66 | 1.00 |
| Antibacterial, Antifungal Functionality | 0.21 | 0.23 |

The results of the building up of the technology roadmap for the eco-friendly building material industry following the three-phased multiple analysis process of patent–Delphi–AHP developed in this research is as illustrated in Figure 6.

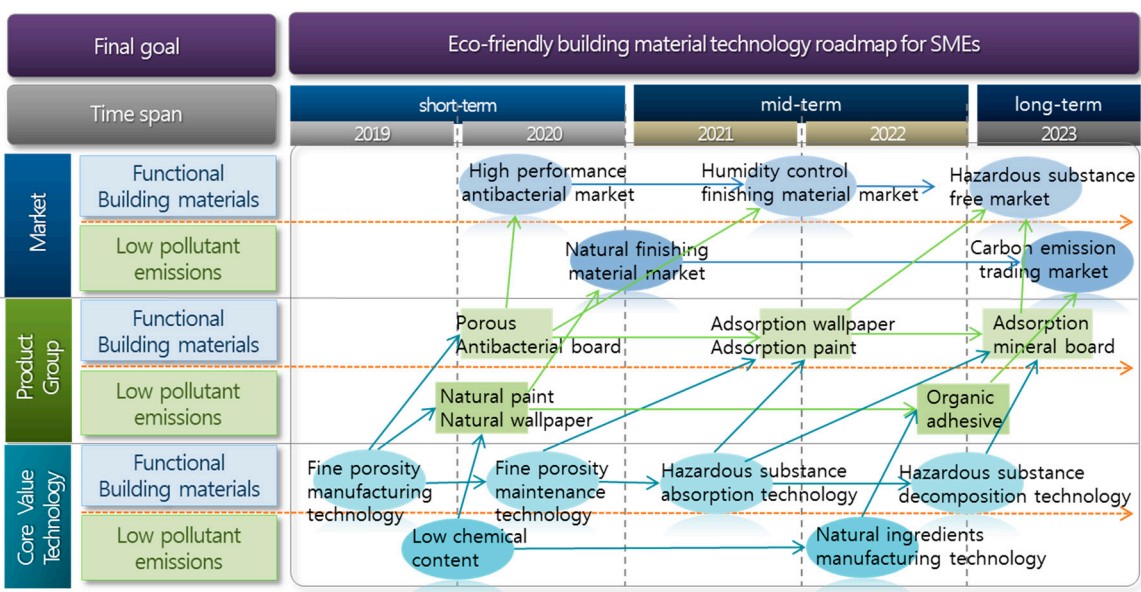

**Figure 6.** Technology roadmap for eco-friendly building materials.

According to Figure 6, the adsorption, moisture absorption, and desorption functional building material technology with the highest importance and highest order of priority in technology development were selected as the most imminent technology subject. Furthermore, research into low-chemical-content technology, which is a low-carbon building material technology, and on the technology for the manufacturing of building materials using natural ingredients should be designed in the mid to long term. With respect to products, a development roadmap for various items of functional building materials and low-carbon building materials that could achieve both low carbon emissions and energy saving must be designed, targeting the hazardous-material-free market in the longer term.

### 4.2. Results of the Verification of Feasibility

Based on the technology roadmap created in Section 4.1, the feasibility was verified by the application of the QFD test. The results of the QFD test are illustrated in Figure 7.

Consequently, it was observed that, among the functions required by customers in the eco-friendly building material market, humidity control demonstrated the highest order of priority, followed by the demand for building materials with resistance to bacteria and fungi. These items could be categorized under adsorption, moisture absorption, and desorption, antibacterial, antifungal building materials,

which confirmed that the development of technologies for these types of functional building materials seem imminent. The results of importance and order of priority analyzed with respect to the correlation between the technology evaluation and the customers' demand demonstrates higher scores for building materials with moisture absorption and desorption and antibacterial and antifungal functions than the adsorption function among functional building materials, which reflects the market demand for the development of building materials that can handle climate changes towards heightened humidity.

The results of the feasibility verification using the QFD test were in accordance with the technology roadmap built up in this research with respect to all of the items—core technology elements, order of priority in the technology development, and importance—which confirms that the results of the eco-friendly building material technology roadmap described in Section 4.1 are feasible.

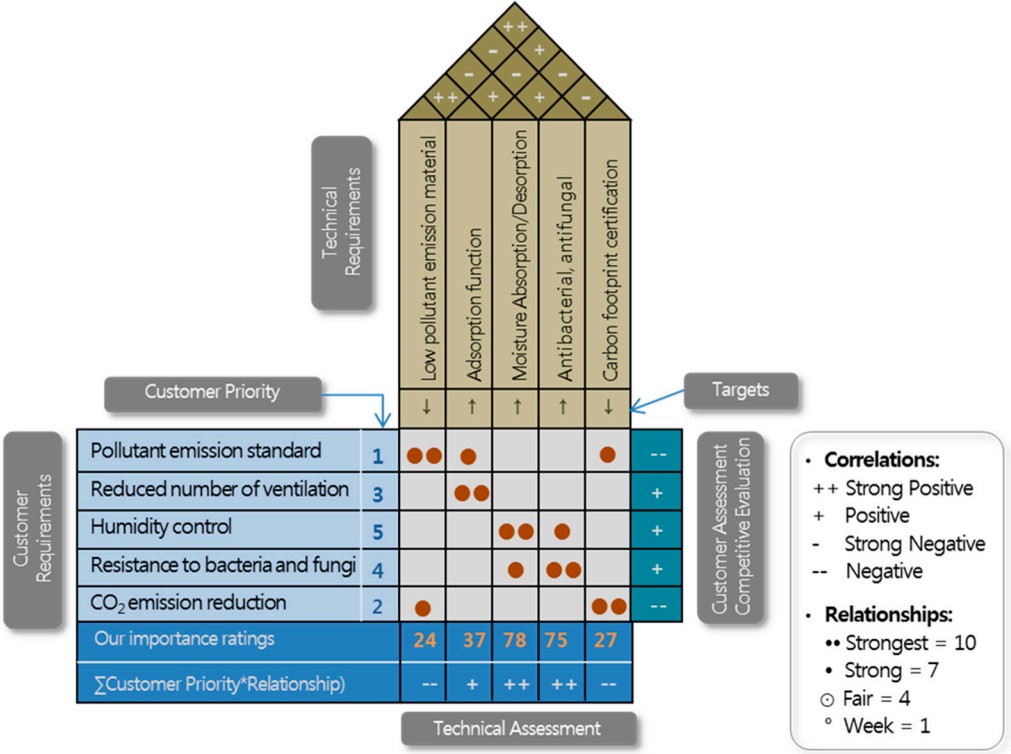

**Figure 7.** QFD test for the eco-friendly building material technology roadmap.

## 5. Discussion and Conclusions

Forecasting future technology is fundamentally flawed, in that it is basically an expectation of future technology, but not an actual definite technology. Therefore, it is more important than anything for industries which lack data, information, and materials on which to base the directions of technology development to set up a proper research method for how to ensure the objectivity and viability of the data that are fundamental to the future forecasting.

This research was intended to build up a technology roadmap to be utilized in the low-energy eco-friendly building material industry within the environmental field, which is facing growing market demand for a countermeasure to recent changes in climate. Since utilization and research of a technology roadmap are not actively in progress in the environmental industry, as it places a higher weight on its social contribution than it does on profit making, despite the fact that active utilization of technology roadmaps can be an important means of grasping technological trends and proposing the direction of technological development in all industries.

As a result of this research, functional building material technologies with adsorption, moisture absorption and desorption were selected as the technical subject of the highest importance and with the highest order of priority for development based on the eco-friendly building material technology roadmap that was built. In this study, eco-friendly technologies for coping with climate change (hot and humidification) were derived and reflected in the roadmap. These technologies include fine-porosity manufacturing technology, fine-porosity maintenance technology, hazardous substance absorption technology, hazardous substance decomposition technology, etc.

Moreover, the verification by QFD test demonstrated that all the core technology elements and the order of technology development priority and importance exhibited identical results to those in the constructed technology roadmap, confirming the feasibility of the research.

This research built up a technology roadmap by deriving a core technology target performance index by performing a multiple analysis process comprising patent analysis–Delphi questionnaire–AHP analysis for the purpose of obtaining objective data, considering the characteristics of the eco-friendly building material industry, which lacks the data and information on the market and future technology; further, this process is different from existing technology roadmap research, because its feasibility is verified by a QFD test, having unlike creativity and differentiation.

Most previous studies have focused mainly on the development direction of the technology itself. These are not just roadmaps that include product development directions based on current market needs, but rather research on methodologies, theories, and principles related to technology trends, technology changes, and prospects and responses. On the other hand, this study can be regarded as the development of multiple roadmaps that present the present problems of corresponding industries in the medium to long term, taking into consideration not only the technical aspects, but also the market needs and product development aspects.

The scope of this study is the case of Korea, where the system and regulations related to functional building materials are actively evolving. Therefore, the Delphi participating policy experts and survey researchers consisted of Koreans. Therefore, it seems that the research results reflect Korea's market situation and Korea's technology, so it is difficult to fully reflect the trends of the global market and the status of technology development. To overcome these limitations, joint research with foreign environmental policy experts representing different contexts, such as developing countries, should be carried out. In addition, this roadmap utilizes key technologies that originate from the key words of the patent, and therefore it is limited with regard to its ability to reflect the latest R&D results (technology before patent application or technology without patent application). Therefore, it will be necessary to study the roadmap methodology in consideration of the technologies proposed in academic research papers, etc.

As the technology roadmap is a process of selecting a core technology that satisfies market demand by forecasting future market trends and developing them into products for business purposes, further R&D capability will have to be nurtured for small and medium companies manufacturing eco-friendly building materials in order to ensure that they have sufficient technological prowess so as to flexibly react to the changes in market trends by themselves, alongside continued screening in future. In addition, the technology roadmap built up in this research is expected to contribute to the improvement of technological competitiveness by being utilized as a core means of appropriate technology development and technology management for small- and medium-sized manufacturing companies in the eco-friendly building material industry in the future.

**Author Contributions:** Conceptualization, H.S. and G.C.; methodology, H.S., T.K. and G.C.; software, H.S. and T.K.; validation, H.S. and G.C.; investigation, T.K. and G.C.; writing-original draft preparation, H.S. and T.K.; writing-review and editing, H.S. and G.C.; supervision, G.C.

**Funding:** This research was supported by a grant (19RERP-B082204-06) from Residential Environment Research Program funded by the Ministry of Land, Infrastructure, and Transport of Korea.

**Conflicts of Interest:** The authors declare no conflict of interest.

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
