# Peer review of "Technology Roadmap for Eco-Friendly Building Materials Industry"

_energies, doi:10.3390/en12050804_

Round 1

Reviewer 1 Report

In the manuscript, authors' build a technology roadmap for the development of technology in the eco-friendly building material industry, using a multiple analysis processes (patent analysis, Delphi, and analytic hierarchy process analysis) and implementing the quality function deployment test to verify the feasibility of the technology roadmap that is constructed.

The topic of the paper is interesting as well as the academic contribution of the work, but the authors should improve the work according to the following indications.

1. In the introduction:

(1.1) where the authors discuss international situation, regulations, and approaches, some recent studies in the field should be considered, among which the following:

“An Evaluation of Energy and Economic Efficiency in Residential Buildings Sector: A Multi-Criteria Analisys on an Italian Case Study”, International Journal of Energy Economics and Policy, 2018, vol. 8(3), pp. 185-196;

(1.2) some sentences should be validated with references (lines 27-31, 32-39, 44-49, 67-81);

(1.3) the authors should explain how the article has been structured by presenting the different sections.

2. Tables and figures should report the sources.

3. The authors should discuss how the results can be interpreted in perspective of previous studies.

4. Limitations and future research directions should be mentioned.

5. Moderate English changes required.

Author Response

1. Introduction:

(1.1) where the authors discuss international situation, regulations, and approaches, some recent studies in the field should be considered, among which the following:

: We have added the recent references which might be useful the research background as indicated.

          (1.2) some sentences should be validated with references (lines 27-31, 32-39, 44-49, 67-81);

             : The appropriate references are presented.

(1.3) the authors should explain how the article has been structured by presenting the different sections.

 The article structure are presented on lines 89-98.

2. Tables and figures should report the sources.

   : The sources are reported on lines 352-354, lines 382-385 for Table1 and 2, respectively.

   : The sources are reported on lines 203-205(Fig.1), lines 308-314(Fig.2), lines 325-328(Fig.3),
     lines 362- 367(Fig.4), lines 372-377(Fig.5), lines 399-401(Fig.6) for Figure 1~6 respectively

3. The authors should discuss how the results can be interpreted in perspective of previous studies.

  : Discussion of the results are added on lines 466-472.

4. Limitations and future research directions should be mentioned.

   : We mentioned the limitations and future research directions on lines 473-482.

5. Moderate English changes required.

              : We have changed appropriately.

Reviewer 2 Report

GENERAL COMMENTS:

The manuscript entitled “Technology Roadmap for Eco-friendly Building Materials Industry” is interesting. The topic addressed by the work fits within the scope of the journal, but also for other Journals of MDPI, such as Buildings. This research seems to be very useful for this field where, in the opinion of the authors, there are no applications that could give objective directions for the future technological development of the building materials industry. After the review, I think it could be a valuable contribution to the Journal.

SPECIFIC COMMENTS:

Abstract: ok

Keywords: ok

Introduction and literature review: the introduction provides sufficient background by showing a specific focus on the situation of the Korean eco-friendly building material industry: market dynamics, government policies, etc. The references mentioned are relevant even if they are not very recent, maybe the bibliography could be updated (if possible).

Methods: The methodology of the technology roadmap applied to the eco-friendly building material industry is described in all phases. The research design is appropriate, using the multiple analysis process to explain and build the technology roadmap. I suggest improving figure 2 and 3: both figures lack the y axis and its value (year); also the x axis is not well done.

Page 7 Line #279 Please delete we observed and change in impersonal form

Page 7 Line #293 Please delete we observed and change in impersonal form

Page 7 Line #301 Please delete we observed and change in impersonal form

Page 10 Line #368 Please delete we observed and change in impersonal form

Results: The results are clearly presented.

Conclusions are supported by the results. Could you add some information about future applications of the TRM?

Author Response

1.     The references mentioned are relevant even if they are not very recent, maybe the bibliography could be updated (if possible).

: We have added the recent references on lines 140-145, 159-163.

2.      I suggest improving figure 2 and 3: both figures lack the y axis and its value (year); also the x axis is not well done.

: Figures 2 and 3 were replaced with new pictures.

3.      Conclusions are supported by the results. Could you add some information about future applications of the TRM?
:
TRM can be extended to TRM considering new technology by using not only patent technology but also academic research papers (lines 479-482).

Reviewer 3 Report

The authors present a research work related to “Technology Roadmap for Eco-friendly Building Materials Industry” where the major contribution is the development of a technology roadmap by deriving core technology target performance index by performing the multiple analysis process of patent analysis-Delphi questionnaire survey-AHP analysis for the purpose of obtaining objective data considering the characteristics of the eco-friendly building material industry which lacks the data and information on the market and future technology.

Remarks to the authors:

1.   In the Literature review section, more recent references can be added.

2.   For the Patent analysis, which is the first phase of the multiple analysis process, please try to extend the period to 2018.

3.   Please review the explanations for Figures 2 and 3. There are contradictions between the main text and figures.

4.   Please argued the obtained results with data which reflects climate changes.

5.   Please use identical and adequate quotation marks in the article.

Author Response

1.  In the Literature review section, more recent references can be added.

   : We have added the recent references on lines 140-145, 159-163.

2.  For the Patent analysis, which is the first phase of the multiple analysis process, please try to extend the period to 2018.

: Figures 2 and 3 were expanded and modified from 2001 to 2017.

3.  Please review the explanations for Figures 2 and 3. There are contradictions between the main text and figures.

  : Some errors were found in the picture description. Therefore, the description has been changed with the new picture. (lines 322-324 and 335-350)

4.  Please argued the obtained results with data which reflects climate changes.

  : We mentioned about it on lines 452-455.

5.  Please use identical and adequate quotation marks in the article.

         : They were modified appropriately and consistently.

Round 2

Reviewer 1 Report

The manuscript has been improved according to indications of my previous review report. Therefore, I suggest its acceptance in present form.

Author Response

We appreciate your valuable comments for the revision

Reviewer 3 Report

1.   Please reconsider the explanations, given at section 4.1. Composition of roadmap (line #311), regarding the considered period of time for the patent analysis, in accordance with the interval mentioned in figures 2 and 3.

“for a 10 year period from 2001 to 2011”

2.   Please do the adequate changes at line #334, where the upper limit of the period taken into consideration, for the number of patent applications of functional building materials, is incorrect.

“increased from 2005 through 2103.”

3.    Please use the impersonal form where it is the case.

For example: We observed that... can be modified with It was observed that...

4.    Please use identical and adequate quotation marks in the article.

For example: It was used ”eco-friendly” line #49 and ‘eco-friendly' line #50.

Author Response

1.   Please reconsider the explanations, given at section 4.1. Composition of roadmap (line #311), regarding the considered period of time for the patent analysis, in accordance with the interval mentioned in figures 2 and 3.

 : Yes, we revised the mistakes as follows:  '17 years', 'from 2001 to 2017. 

2.   Please do the adequate changes at line #334, where the upper limit of the period taken into consideration, for the number of patent applications of functional building materials, is incorrect.

“increased from 2005 through 2103.

: The descriptions of Fig. 2 &3 have been changed precisely (Lines 321-324, 340-346). 

3.    Please use the impersonal form where it is the case.

For example: We observed that... can be modified with It was observed that...

: We have modified those representations as indicated.

4.    Please use identical and adequate quotation marks in the article.

For example: It was used ”eco-friendly” line #49 and ‘eco-friendly' line #50.

 : We  fixed them appropriately throughout the manuscript.